# 16q24.3 Microdeletions Disrupting Upstream Non-Coding Region of *ANKRD11* Cause KBG Syndrome

**DOI:** 10.3390/genes16020136

**Published:** 2025-01-24

**Authors:** Aiko Iwata-Otsubo, Alyssa L. Rippert, Jorune Balciuniene, Sarah K. Fiordaliso, Robert Chen, Preetha Markose, Cara M. Skraban, Christopher Gray, Elaine H. Zackai, Holly A Dubbs, Matthew A. Deardorff, Laura K. Conlin, Kosuke Izumi

**Affiliations:** 1Division of Human Genetics, Department of Pediatrics, The Children’s Hospital of Philadelphia, Philadelphia, PA 19104, USA; 2Division of Genomic Diagnostics, Department of Pathology and Laboratory Medicine, The Children’s Hospital of Philadelphia, Philadelphia, PA 19104, USA; 3Roberts Individualized Medical Genetics Center, Department of Pediatrics, The Children’s Hospital of Philadelphia, Philadelphia, PA 19104, USA; 4Department of Pediatrics, Perelman School of Medicine, University of Pennsylvania, Philadelphia, PA 19104, USA; 5Department of Pathology and Laboratory Medicine, Perelman School of Medicine, University of Pennsylvania, Philadelphia, PA 19104, USA

**Keywords:** *ANKRD11*, developmental delay, KBG syndrome, transcriptome, molecular diagnosis, gene expression

## Abstract

**Background:** KBG syndrome is a multisystem developmental disorder characterized by macrodontia of the upper permanent incisors, distinctive facial features, a short stature, developmental delay, variable intellectual disability, and behavioral issues. Heterozygous chromosomal deletion encompassing the partial or entire *ANKRD11* gene, as well as the loss of function mutations, result in haploinsufficiency of the gene, leading to KBG syndrome. This indicates that precise levels of *ANKRD11* transcripts or protein are essential for human development. **Clinical report:** Here, we report three individuals who present with clinical features of KBG syndrome. These individuals carry microdeletions encompassing only the non-coding exon 1 of *ANKRD11* and its upstream region. Our molecular analysis showed that this deletion leads to reduction in the *ANKRD11* transcript and global transcriptome alterations similar to those seen in KBG syndrome patients. **Conclusions:** We concluded that microdeletions involving non-coding exon 1 of *ANKRD11* lead to KBG syndrome. Our study suggests the utility of transcriptome analysis in aiding the interpretation of novel copy number variants in the non-coding genomic region of *ANKRD11*.

## 1. Introduction

*ANKRD11*, encoding a member of the family of ankyrin repeat-containing cofactors, is implicated in neuron differentiation during brain development [1]. Loss-of-function mutations in *ANKRD11* have been associated with a wide spectrum of clinical phenotypes, including KBG syndrome, as well as other presentations such as Coffin–Siris-like syndrome and intellectual disabilities with infantile spasms [2,3]. KBG syndrome is characterized by developmental delay, variable intellectual disability, behavioral issues, a short stature, macrodontia of the upper permanent incisors, and distinctive facial features, including a triangular face, short neck, wide or bushy eyebrows, synophrys, a prominent nasal bridge, hypertelorism, prominent ears, a long and smooth philtrum and a thin upper lip [4,5,6,7]. Other features include feeding difficulties, brachydactyly, scoliosis, hearing loss, seizures, and brain malformations.

Most *ANKRD11* mutations arise de novo; however, familial segregation in an autosomal dominant manner has also been described [4]. The majority of *ANKRD11* mutations associated with KBG syndrome are nonsense, frameshift and splicing mutations, in addition to chromosomal deletions involving the full or partial sequence of the gene, all of which could result in a reduction in the *ANKRD11* transcript level. Therefore, maintaining the precise transcript level of *ANKRD11* is essential during human development. The observation of a reduction in the *ANKRD11* transcript level could be used as a diagnostic marker, and the demonstration of a lower *ANKRD11* transcript level may support the possibility of the identified variant being pathogenic when haploinsufficiency is suspected.

ANKRD11 is known as a transcriptional regulator that interacts with p160 coactivator and histone deacetylases (HDACs), inhibiting ligand-dependent transcriptional activation [8]. It has been shown that ANKRD11 functions as a nuclear co-regulator in the developing brain by regulating histone acetylation and gene expression. Yoda mice carrying a point mutation in the *Ankrd11* HDAC-interacting domain exhibited a significant number of differentially expressed genes, ~60% of which were downregulated [9]. It is highly possible that a reduction in functional ANKRD11 causes global transcriptional alterations in humans. Analyzing global transcriptome changes in patient samples could provide insights into ambiguous variants in the *ANKRD11* gene and its adjacent genomic regions.

Here, we report three individuals from two families with microdeletions involving *ANKRD11* non-coding exon 1 and the sequence upstream of the gene. Performing transcriptome analysis via RNA-sequencing, we demonstrate the pathogenicity of this microdeletion.

## 2. Clinical Report

### 2.1. Case 1 and 2

Case 1 is a 3-year-old female with developmental delay. Her prenatal history is remarkable for maternal alcohol use. Case 1 was born at 40 weeks, weighing 3.16 kg (25th–50th percentile). Her APGAR scores were 7 at 1 min and 9 at 5 min, and she had an unremarkable neonatal course. Concerns were originally raised due to developmental delay. She started sitting at 10 months and walking at 17 months. She had significant speech delay, with no first words until 2 years of age and requiring speech therapy weekly. At 5 years old, she has deficits in her functional use of language and social interactions. Case 1 had an unremarkable brain MRI at 6 months of age. Her weight gain has been slow, and she has a short stature. The family history of Case 1 is limited as she is adopted, but some of her biological siblings are similarly affected by developmental delay. She has not experienced any suspicious seizure-like episodes. Hence, she has not had an electroencephalogram (EEG). She has not experienced any clinical symptoms suggestive of olfactory dysfunction.

At 3 years and 1 month, her height was 92.6 cm (28th percentile), her weight was 12.7 kg (18th percentile), and her head circumference was 48.5 cm (46th percentile). Her physical examination revealed a triangular face, arched eyebrows, upslanted palpebral fissures, normal upper frontal central incisors, 5th finger clinodactyly, and brachydactyly, and these features do not reflect fetal alcohol syndrome (Figure 1). Chromosome SNP microarray analysis revealed the heterozygous deletion of a 32 kb region within chromosome 16q24.3 involving *ANKRD11* and *SPG7* (arr[hg19] 16q24.3(89,544,636-89,577,046)x1) (Figure 2). Her whole-exome sequencing test was negative.

Her similarly affected biological sister (Case 2) was also found to carry the same heterozygous 16q24.3 deletion. The clinical symptoms of Case 2 include early developmental delay, learning differences, behavioral differences, bilateral conductive hearing loss, microcephaly, and feeding difficulties. Her physical features are remarkable, with a low anterior hairline, full brows with mild synophrys, a broad nasal bridge and tip, and prominent upper central incisors. She has not experienced any suspicious seizure-like episodes. She has not experienced any clinical symptoms suggestive of olfactory dysfunction. She has not had brain MRI or EEG.

### 2.2. Case 3

Case 3 is an 11-year-old male with developmental delay, pulmonic stenosis and ventricular septal defect (VSD). He was born after 40 weeks of gestation, and upon birth, he was found to have a VSD requiring surgical repair. At 19 months, Case 3 presented with complex partial seizures, which were localized to a region of intraparenchymal hemorrhage and thought to be traumatic in nature. Brain MRI revealed mild hypogenesis of the inferior vermis; otherwise, his brain structure was unremarkable. Case 3 had right cryptorchidism and an inguinal hernia, both of which have been repaired. Other medical problems include cholelithiasis, which was surgically treated. He has a history of development delay. His first meaningful word was spoken when he was 4 years old. His recent neuropsychological evaluation identified an IQ of 65, consistent with mild intellectual disability. He sat after 10 months, crawled at 18 months, and walked by 22 months. Case 3 had a history of seizure-like episodes, which were thought to be related to a non-accidental trauma. He had a mildly abnormal EEG due to left posterior quadrant amplitude suppression of uncertain significance. He has not experienced any clinical symptoms suggestive of olfactory dysfunction. His family history is limited as he is adopted.

At 11 years and 4 months, his height was in the 22nd percentile, his weight was in the 54th percentile, and his head circumference was in the 75th percentile. Physical examination revealed a triangular face, hypertelorism (direct interpupillary distance of 6 cm, 90th percentile), upslanted palpebral fissures, large ears (6.5 cm, +2SD for age), large upper front teeth, a high arched palate, brachydactyly, bilateral 5th finger clinodactyly and broad terminal phalanges. He did not have gross motor concerns at the time of this evaluation. His genetic testing included SNP microarray, Fragile X syndrome testing, and Noonan syndrome panel, all with negative results. SNP array analysis revealed a heterozygous deletion of a 25 kb region within chromosome 16q24.3 involving *ANKRD11* and *SPG7* (arr[hg19] 16q24.3(89,549,562-89,574,576)x1) (Figure 2). The deletion was confirmed by droplet digital PCR (ddPCR) that was based on TaqMan probe assay targeting a non-coding exon 1 of *ANRKD11*. A reference assay for *RPP30* gene was run in the same reaction and was used to compute a relative copy number of the *ANKRD11* target region.

## 3. Materials and Methods

Lymphoblastoid cell lines (LCLs) were established from Case 1, two other patients diagnosed with KBG syndrome and three healthy control subjects. The *ANKRD11* variants present in the two additional KBG syndrome patients were a heterozygous frameshift mutation (c.2814_2818del; p.Asp938Glufs*78) and heterozygous intragenic deletion of exons 4-11, respectively. LCLs were cultured in RPMI 1640 with 300 mg/L L-glutamine (Life Technologies, 11875085, Carlsbad, CA, USA) supplemented with 20% HyClone FBS (Fisher Scientific, SH3007103, Waltham, MA, USA), 0.2% penicillin-streptomycin (Life Technologies, 15140122, Carlsbad, CA, USA), 0.2% Plasmocin (InvivoGen, ant-mpp, San Diego, CA, USA), and 1% Glutamax (Life Technologies, 35050061, Carlsbad, CA, USA).

Total RNA samples were extracted using TRIzol (Invitrogen, 15-596-018, Waltham, MA, USA) and Nucleospin RNA (Macherey-Nagel, 740955.50, Allentown, PA, USA) following the manufacturer’s instructions, as previously described [10]. The extracted RNA samples were used for RNA sequencing and quantitative RT-PCR (qRT-PCR). RNA was reverse transcribed with TaqMan Reverse Transcription Reagents (Applied Biosystems, N8080234, Waltham, MA, USA). Synthesized cDNA was used for quantitative PCR analysis using Fast SYBR™ Green Master Mix (Applied Biosystems, 4309155, Waltham, MA, USA) on an ABI 7500 PCR system (Applied Biosystems, Waltham, MA, USA). The gene expression levels of *ANKRD11* were normalized against *GAPDH* expression levels. The following primer sequences were used for qPCR: *ANKRD11* (FW: 5′-CTCGACGGAGAGCTCAGAAG-3′, RV: 5′-CATACTCGTCCTTGACGGGG-3′) and *GAPDH* (FW: 5′-GCACCGTCAAGGCTGAGAAC-3′, RV: 5′-TGGTGAAGACGCCAGTGGA-3′).

RNA sequencing was performed by Novogene (Sacramento, CA, USA). About 40–50 M raw reads were generated by HiSeq, with paired-end 150 bp sequencing for each sample. The sequenced reads were mapped to the GRCh38 reference genome. Data were analyzed by a CLC genomics workbench with default analysis settings (https://digitalinsights.qiagen.com/products-overview/discovery-insights-portfolio/analysis-and-visualization/qiagen-clc-genomics-workbench/) (accessed on 14 January 2025). The reads were mapped to coding genes and ncRNA genes. The mapping statistics can be found in Appendix A.

## 4. Results

RNA-seq revealed that the entire *ANKRD11* transcript was decreased in the LCLs of Case 1 compared to the female control samples (Figure 3A). The *ANKRD11* expression level was 1.92-fold decreased in the Case 1 sample compared to the three control samples (*p*-value −1.65 × 10^−5^, FDR 2.71 × 10^−3^). A reduction in *ANKRD11* gene expression was confirmed by qPCR (Figure 3B), and these results suggest the haploinsufficiency of *ANKRD11* as an underlying cause for this patient’s clinical features.

Since *ANKRD11* is a transcription regulator, we hypothesized that the disruption of *ANKRD11* likely causes global transcriptional alteration. Using two male LCLs established from the patients with KBG syndrome carrying intragenic pathogenic *ANKRD11* variants, and two male control LCLs, we identified differentially expressed genes (DEGs) (total 628 genes: 217 upregulated and 411 downregulated) in LCLs carrying pathogenic *ANKRD11* variants (FDR < 0.05) (Appendix A). DEGs are predominantly downregulated in these two patients with pathogenic *ANKRD11* variants, which is similar to observations in the *Ankrd11* mutant mouse model [9]. Next, we identified DEGs in the LCLs of Case 1 compared to male control LCLs. The DEGs are predominantly downregulated (total 547 genes: 108 upregulated and 439 downregulated) in the LCLs of Case 1 (Appendix A). We further questioned whether upregulated and downregulated genes in KBG syndrome patients with pathogenic *ANKRD11* variants overlap with those in Case 1. The comparison indicated that a majority of downregulated genes in patients with pathogenic *ANKRD11* variants were also downregulated in the LCLs of Case 1, while the upregulated genes did not show significant overlap (Figure 4). Since DEGs are predominantly downregulated in both KBG syndrome and Case 1, and the majority of the downregulated genes overlap between the two, we concluded that the microdeletion seen in Case 1 causes global transcriptional alterations similar to those observed in patients with intragenic pathogenic *ANKRD11* mutations.

## 5. Discussion

Here, we report three individuals with microdeletions in 16q24.3 involving the upstream region and exon 1 in a 5′ untranslated region (5′ UTR) of *ANKRD11*. Our clinical evaluation indicates that these patients demonstrate the clinical features seen in KBG syndrome, including a triangular face, 5th finger clinodactyly and brachydactyly, and developmental delay. Case 2 and 3 showed some additional features that overlap with those seen in KBG syndromes, including prominent upper central incisors and hypertelorism [7]. Deletions encompassing only exon 1 of the gene have been recently described in individuals with KBG syndrome [11,12,13]. Collectively, we concluded that the microdeletion spanning the upstream region and 5′UTR of *ANKRD11* causes KBG syndrome.

Deletions in this chromosomal region have been associated with 16q24.3 microdeletion syndrome that ranges from 137 kb to 2 Mb and include *ANKRD11* as well as other flanking genes [14]. Microdeletion syndromes including 16p24.3 microdeletion syndrome are the genetic diagnoses caused by chromosomal deletions. Individuals with 16q24.3 microdeletion syndrome have features seen in KBG syndrome and additional features, including autism spectrum disorder, brain anomalies, congenital heart defects, severe astigmatism, and thrombocytopenia [14,15]. A comparison of these deletions suggested potential critical genes such as *ZNF778*, *ZFPM1* and *CDH15,* in addition to *ANKRD11*, which differ from our cases. The deletions in our patients are smaller than those seen in individuals with microdeletion syndrome and involve only *ANKRD11* non-coding 1 and partial SPG7. Deletions in the three patients also include at least the first exon of the *SPG7* gene, which is associated with autosomal recessive spastic paraplegia 7. Although one individual carrying a heterozygous nonsense variant in the gene was reported to have the condition, heterozygous carriers are typically asymptomatic [14,16]. Therefore, it is unlikely that the partial deletion of the *SPG7* gene contributes to patients’ phenotypes.

Our transcriptome analysis showed a reduction in *ANKRD11* expression as a consequence of *ANKRD11* 5′UTR deletion in the LCL sample of Case 1. This observation is consistent with the findings of Bestetti et al. [11] and Borja et al. [12], which showed that an *ANKRD11* exon 1 deletion caused reduced gene expression when performing qRT-PCR in the patient blood and saliva samples. Additionally, we found global transcriptional alterations similar to those observed in patients with pathogenic *ANKRD11* variants, indicating that *ANKRD11* 5′UTR deletion has comparable effects to known *ANKRD11* pathogenic variants. The primary role of the 5′ UTR is in the regulation of protein translation [17]. Therefore, the deletion of the 5′ UTR alone represents an unlikely mechanism of *ANKRD11* transcript reduction. The microdeletion spans the entire intragenic region between *ANKRD11* and *SPG7*, including upstream genetic elements such as promoters or other cis-acting elements of *ANKRD11*. The histone H3K27Ac peaks obtained in the ENCODE project were found within this intragenic region, indicating the presence of active promoters or enhancers within this microdeletion [18]. Thus, the disruption of upstream transcriptional regulatory elements, such as promoter or enhancer elements, is more likely the cause of reduced *ANKRD11* expression.

In summary, we report three individuals presenting with KBG syndrome due to the deletion of non-coding exon 1 of *ANKRD11*. Our findings underscore the importance of the genomic evaluation of non-coding regions of *ANKRD11* in individuals with suspected KBG syndrome and suggest the utility of transcriptome analysis in establishing a molecular diagnosis of KBG syndrome.

## Figures and Tables

**Figure 1 genes-16-00136-f001:**
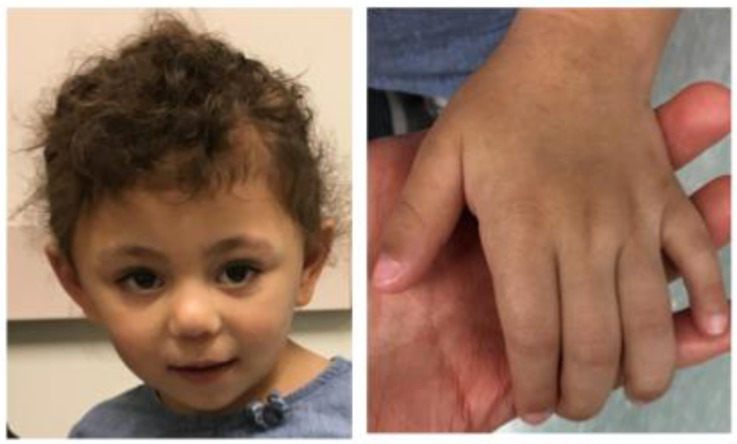
Physical features of an individual with an *ANKRD11* non-coding region deletion. Physical features of Case 1 individual, showing a triangular face, arched eyebrow, upslanted palpebral fissures, normal upper frontal central incisors, 5th finger clinodactyly and brachydactyly.

**Figure 2 genes-16-00136-f002:**
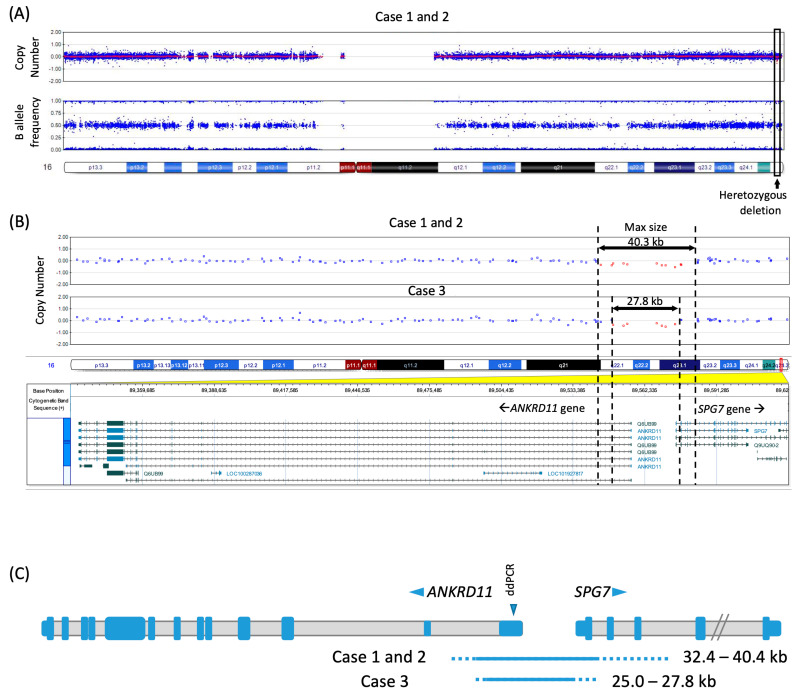
SNP array data identified a heterozygous microdeletion affecting non-coding exon 1 and the sequence upstream of the *ANKRD11* gene. (**A**) SNP array analysis revealed a short heterozygous deletion in the 16q24.3 region. (**B**) The deletion encompasses 11 SNPs for Case 1/2 and 7 SNPs for Case 3, with a maximum deleted size of 40.3 kb and 27.8 kb, respectively. (**C**) Schematic representation of *ANKRD11* and *SPG7* genes affected by the deletions. *ANKRD11* and *SPG7* are transcribed in opposite orientation, as denoted by horizontal arrowheads. Exonic sequences are indicated by blue boxes, with coding exons depicted taller than the non-coding ones. All 13 exons of the *ANKRD11* (NM_013475) gene and 5 out of 17 exons of the *SPG7* gene (NM_003119) are shown. A contiguous line denotes the minimum deletion region demarcated by the deleted SNPs, while a dashed line extends the region to the next non-deleted SNP and therefore incorporates the breakpoint. The numbers next to the lines indicate the minimum and maximum sizes of the deletions in each patient. A vertical arrowhead denotes the region in *ANKRD11* exon 1 that was tested by ddPCR in Case 3 to confirm the deletion called by the SNP array testing.

**Figure 3 genes-16-00136-f003:**
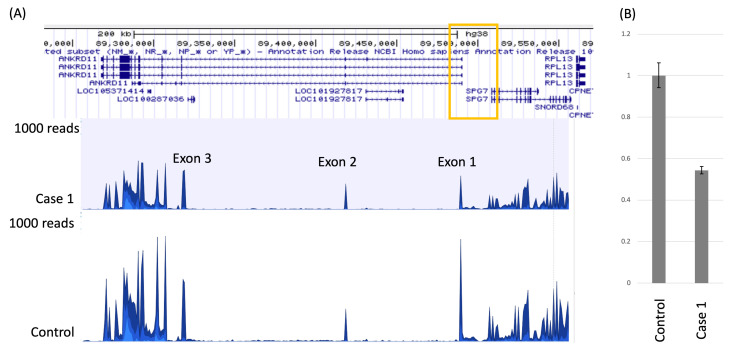
Reduction in *ANKRD11* transcript due to the deletion of the *ANKRD11* non-coding region. (**A**) A reduction in the *ANKRD11* transcript was identified by RNA-seq. The chromosomal deletion seen in Case 1 is depicted by a yellow square in the genome browser view of the *ANKRD11* locus. The Y axis represents the read number of the genomic position. Although the deletion spans exon 1 of the *ANKRD11*, the transcript level of the entire *ANKRD11* is reduced in the Case 1 sample compared to the female control samples. (**B**) A reduction in the *ANKRD11* transcript was confirmed by qPCR. Representative data of RT-qPCR showing the reduction in the *ANKRD11* transcript in the patient LCLs compared to the female control LCLs. RT-qPCR was conducted in triplicate and twice independently. *GAPDH* was used as an endogenous control to normalize the total amount of cDNA in the control and patient LCLs. Error bars, 2SD.

**Figure 4 genes-16-00136-f004:**
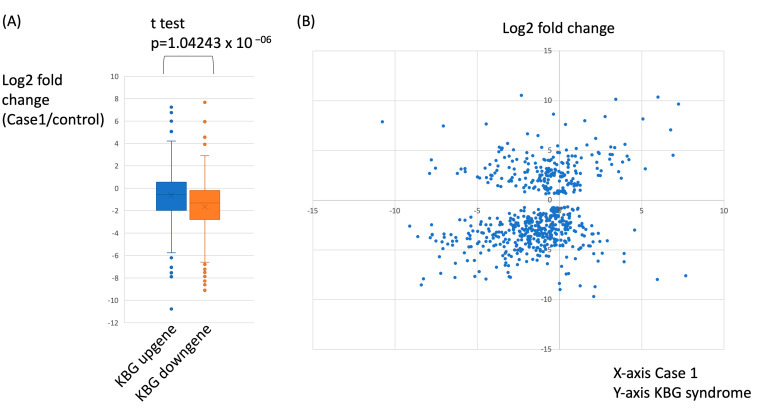
Comparison of differentially expressed genes in the proband and other known KBG cases. (**A**) Boxplot shows a comparison of differentially expressed genes (DEGs) between patients with KBG syndrome and the Case 1 patient. Using RNA sequencing data of the known KBG cases, we identified DEGs that include upregulated and downregulated genes in patients with KBG syndrome. The *Y*-axis shows the log2-fold differences in gene expression between the control and Case 1. The majority of downregulated genes in KBG syndrome are also downregulated in Case 1. (**B**) Scatter plot demonstrating the correlation between the differentially expressed genes (DEGs) of Case 1 and known KBG syndrome cases. The *X*-axis represents the log2-fold differences in gene expression between Case 1 and the controls. The *Y*-axis represents the log2-fold differences in gene expression between the known KBG cases and the controls. The majority of the DEGs clustered in the left lower quadrant, indicating that similar genes were downregulated in Case 1 and other known KBG syndrome cases.

## Data Availability

The original contributions presented in this study are included in the article/Appendix A. Further inquiries can be directed to the corresponding authors.

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
