# Peer review of "16q24.3 Microdeletions Disrupting Upstream Non-Coding Region of ANKRD11 Cause KBG Syndrome"

_genes, 2025, doi:10.3390/genes16020136_

Round 1
Reviewer 1 Report
Comments and Suggestions for Authors
To the Authors
ANKRD11 encoding a member of the family of ankyrin repeat-containing cofactors is implicated to play a role in neuron differentiation during brain development. Loss-of-function (LOF) mutations in ANKRD11 have been associated with a wide spectrum of clinical phenotypes including KBG syndrome as well as other presentations such as Coffin-Siris-like syndrome, and intellectual disabilities with infantile spasms. The Authors report three case-reports presenting with features of KGB syndrome carrying microdeletions encompassing only the non-coding exon 1 of ANKRD11 and its upstream region (i.e., microdeletions in 16q24.3 involving the upstream region and exon 1 in a 5’ untranslated region (5’ UTR) of ANKRD11). Deletions in the three patients also included at least the first exon of the SPG7 gene, which is associated with autosomal recessive spastic paraplegia 7. Despite one individual carrying a heterozygous nonsense variant in the gene has been reported to have the condition, heterozygous carriers are typically asymptomatic and the Authors reasoned that it is unlikely that partial deletion of the SPG7 gene contributes to patients’ phenotypes. Molecular analyses indicated a reduction of the ANKRD11 transcript and global transcriptome alterations similar to those seen in KBG syndrome patients. The AAs conclude that microdeletions involving the non-coding exon 1 of ANKRD11 lead to KBG syndrome.
The study is interesting and adds further knowledge to the KGB syndrome. Nonetheless, a few minor points should be addressed.
Minor Points
1. Case 1. I would suggest also including the corresponding birth weight in the metric system measures (oz to kg).
2. Case number presentation is quite confusing: 3 or 2? We do understand that case 1 is an adopted girl, case 2 is her biological sister and case 3 is a genetically unrelated male. However, some inconsistencies in the text are apparent (Abstract: “Here, we report three individuals who present with clinical features of KBG syndrome”; Introduction: “Here we report three individuals from two families with microdeletions involving ANKRD11 non-coding exon 1 and the sequence upstream of the gene”; Fig 2 is relating to three cases (case 1 and 2; case 3); Fig 3 is relating to Case 1 vs control; Fig 4 apparently relates to case 1 (“the proband”); Discussion: “Here, we report three individuals with microdeletions in 16q24.3 involving the upstream region and exon 1 in a 5’ untranslated region (5’ UTR) of ANKRD11”; In summary, we report two individuals presenting with KBG syndrome due to the deletion of non-coding exon 1 of ANKRD11). Please correct and/or clarify.
3. Brain anatomy findings. Brain MRI is reported for Case 3 only (“Brain MRI revealed mild hypogenesis of the inferior vermis; otherwise, his brain structure was unremarkable”). Was brain imaging available for cases 1 / 2? Please add or clarify.
4. Seizures: Seizures are frequently described in the KBG spectrum phenotype. Apparently none of the three mentioned cases experienced epilepsy. Did the cases undergo an EEG recording? Please clarify.
5. Olfactory bulb anomalies have been recently reported in KBG sdr mouse models and patients (Goodkey K, et al. Olfactory bulb anomalies in KBG syndrome mouse models and patients. BMC Med. 2024 Apr 15;22(1):158. doi: 10.1186/s12916-024-03363-6). Were any of the patients with microdeletions involving the non-coding exon 1 of ANKRD11 tested for olfactory anomalies? Please clarify.
Author Response
Point 1: Case 1. I would suggest also including the corresponding birth weight in the metric system measures (oz to kg).
Response: Per the suggestion of the reviewer 1, we changed it the number in the metric system.
Point 2: Case number presentation is quite confusing: 3 or 2? We do understand that case 1 is an adopted girl, case 2 is her biological sister and case 3 is a genetically unrelated male. However, some inconsistencies in the text are apparent (Abstract: “Here, we report three individuals who present with clinical features of KBG syndrome”; Introduction: “Here we report three individuals from two families with microdeletions involving ANKRD11 non-coding exon 1 and the sequence upstream of the gene”; Fig 2 is relating to three cases (case 1 and 2; case 3); Fig 3 is relating to Case 1 vs control; Fig 4 apparently relates to case 1 (“the proband”); Discussion: “Here, we report three individuals with microdeletions in 16q24.3 involving the upstream region and exon 1 in a 5’ untranslated region (5’ UTR) of ANKRD11”; In summary, we report two individuals presenting with KBG syndrome due to the deletion of non-coding exon 1 of ANKRD11). Please correct and/or clarify.
Response: Our apologies for the error. We corrected the error in the revised manuscript.
Point 3: Brain anatomy findings. Brain MRI is reported for Case 3 only (“Brain MRI revealed mild hypogenesis of the inferior vermis; otherwise, his brain structure was unremarkable”). Was brain imaging available for cases 1 / 2? Please add or clarify.
Response: Case 1 had an unremarkable brain MRI, but Case 2 did not have any imaging studies. We added this information in the revised manuscript.
Point 4: Seizures: Seizures are frequently described in the KBG spectrum phenotype. Apparently none of the three mentioned cases experienced epilepsy. Did the cases undergo an EEG recording? Please clarify.
Response: While Case 1 and Case 2 have not had any seizures, Case 3 had history of seizure like episodes, which were thought to be related to a non-accidental trauma. He had mildly abnormal EEG due to left posterior quadrant amplitude suppression of uncertain significance. We added this information in the revised manuscript.
Point 5: Olfactory bulb anomalies have been recently reported in KBG sdr mouse models and patients (Goodkey K, et al. Olfactory bulb anomalies in KBG syndrome mouse models and patients. BMC Med. 2024 Apr 15;22(1):158. doi: 10.1186/s12916-024-03363-6). Were any of the patients with microdeletions involving the non-coding exon 1 of ANKRD11 tested for olfactory anomalies? Please clarify.
Response: None of the cases showed clinical signs suggestive of olfactory anomalies. We added this information in the revised manuscript.
Reviewer 2 Report
Comments and Suggestions for Authors
Dear Author,
I have reviewed significant case reports on KBG syndrome. With the recent introduction of panel testing for rare diseases, I believe that more diagnoses of this disorder will be made in the future. I also believe that it will become one of the rarest diseases to be recognized accordingly. I will list my comments on this case report below.
Major 1: Please provide an explanation of the microdeletion syndrome.
Minor 1: Exon 1 is mainly mentioned, but there are scattered papers on exon 9 in the existing reports. How would you interpret this difference?
Minor 2: Radiological findings in KBG syndrome include reports of cerebellar hypoplasia. How was the MRI in the presenting case?
Minor 3: Short finger syndrome is also known in Coffin-Siris syndrome. Is there any association?
Minor 4: If there are any clinical diagnostic criteria for KBG syndrome, please let me know.
Minor 5: You mention that observation of decreased transcription level of ANKRD11 is a diagnostic marker for suspected haploinsufficiency, please be more specific and clear.
Minor 6: As mentioned in references 15 and 16, does 16q24.3 contain other candidate disease genes besides ANKRD11? Please explain as much as you know.
Minor 7: Please add a reference to the clinical management and treatment of KBG syndrome.
Thank you for your contribution to this issue of "genes". I look forward to seeing your revised manuscript.
Best regards,
Dr. Reviewer
Author Response
Major 1: Please provide an explanation of the microdeletion syndrome.
Response: Microdeletion syndromes are the genetic diagnoses caused by chromosomal deletions. We added this explanation in the revised text.
Minor 1: Exon 1 is mainly mentioned, but there are scattered papers on exon 9 in the existing reports. How would you interpret this difference?
Response: Exon 1 is a part of the untranslated region, meanwhile exon 9 comprises the coding region of ANKRD11 protein.
Minor 2: Radiological findings in KBG syndrome include reports of cerebellar hypoplasia. How was the MRI in the presenting case?
Response: Brain MRI was performed in the Case 1 and Case 3. While Case 3 had hypogenesis of the inferior vermis, which is a type of cerebellar hypoplasia, Case 1 had normal brain MRI.
Minor 3: Short finger syndrome is also known in Coffin-Siris syndrome. Is there any association?
Response: Short fingers are a non-specific finding, that can be found not only in Coffin-Siris syndrome, but also in many other developmental disorders including KBG syndrome.
Minor 4: If there are any clinical diagnostic criteria for KBG syndrome, please let me know.
Response: Several papers have proposed clinical diagnostic criteria (Brancati et al, Skjei et al, and Low et al.), but no consensus diagnostic criteria have been established.
- Brancati F, Sarkozy A, Dallapiccola B (2006). KBG syndrome. Orphanet J Rare Dis. 12;1:50.
- Skjei KL, Martin M, Slavotinek A. KBG syndrome: report of twins, neurological characteristics, and delineation of diagnostic criteria. Am J Med Genet A. 2007;143A:292–300.
- Low K, Ashraf T, Canham N, Clayton-Smith J, Deshpande C, Donaldson A, Fisher R, Flinter F, Foulds N, Fryer A, et al. Clinical and genetic aspects of KBG syndrome. Am J Med Genet A. 2016;170:2835–46.
Minor 5: You mention that observation of decreased transcription level of ANKRD11 is a diagnostic marker for suspected haploinsufficiency, please be more specific and clear.
Response: We modified the sentence as follows: The observation of a reduction in ANKRD11 transcript level could be used as a diagnostic marker and the demonstration of a lower ANKRD11 transcript level may support the possibility of the identified variant being pathogenic when haploinsufficiency is suspected.
Minor 6: As mentioned in references 15 and 16, does 16q24.3 contain other candidate disease genes besides ANKRD11? Please explain as much as you know.
Response: The chromosomal microdeletion seen in our patients involved ANKRD11 and SPG7, and haploinsufficiency of SPG7 is not thought to be associated with human disorders. Therefore, ANKRD11 is the only gene explaining the clinical symptoms seen in our patients.
Minor 7: Please add a reference to the clinical management and treatment of KBG syndrome.
Response: Clinical management and treatment are based on the symptoms, and there are no specific treatment towards KBG syndrome.
Round 2
Reviewer 2 Report
Comments and Suggestions for Authors
Dear Authors,
The authors responded appropriately to all major and minor questions from the reviewers. The category of publication is ”Case Report”. I support the publication of this article.
Best regards, Dr. Reviewer